# Morphological and Phylogenetic Characterization Reveals Five New Species of *Samsoniella* (Cordycipitaceae, Hypocreales)

**DOI:** 10.3390/jof8070747

**Published:** 2022-07-19

**Authors:** Zhiqin Wang, Yao Wang, Quanying Dong, Qi Fan, Van-Minh Dao, Hong Yu

**Affiliations:** 1Yunnan Herbal Laboratory, College of Ecology and Environmental Sciences, Yunnan University, Kunming 650091, China; W18314560773@163.com (Z.W.); wangyao1@aliyun.com (Y.W.); nympheel@163.com (Q.D.); 2The International Joint Research Center for Sustainable Utilization of Cordyceps Bioresources in China and Southeast Asia, Yunnan University, Kunming 650091, China; 15925137267@139.com; 3Institute of Regional Research and Development, Ministry of Science and Technology, Hanoi 10055, Vietnam; minhdaovan87@gmail.com

**Keywords:** isaria-like fungi, micromorphology, phylogenetic analyses, taxonomy

## Abstract

*Samsoniella* is a very important fungal resource, with some species in the genus having great medical, economic and ecological value. This study reports five new species of *Samsoniella* from Yunnan Province and Guizhou Province in Southwestern China and Dole Province in Vietnam, providing morphological descriptions, illustrations, phylogenetic placements, associated hosts and comparisons with allied taxa. Based on morphological observations and phylogenetic analyses of combined nr*SSU*, nr*LSU*, *tef-1α*, *rpb1* and *rpb2* sequence data, it was determined that these five new species were located in the clade of *Samsoniella* and different from other species of *Samsoniella*. The five novel species had morphologies similar to those of other species in the genus, with bright orange cylindrical to clavate stromata (gregarious). The fertile part lateral sides usually had a longitudinal ditch without producing perithecia, and superficial perithecia. The phialides had a swollen basal portion, tapering abruptly into a narrow neck and oval or fusiform one-celled conidia, often in chains. The morphological characteristics of 23 species in *Samsoniella*, including five novel species and 18 known taxa, were also compared in the present study.

## 1. Introduction

*Samsoniella* was established by Mongkolsamrit et al. (2018) based on morphological and molecular evidence to accommodate three isaria-like species: the type species *S. inthanonensis* and two other species, *S. alboaurantia* and *S. aurantia* [1]. *Samsoniella* species were characterized by the formation of bright orange, cylindrical to clavate stromata (gregarious). The fertile part lateral sides usually had a longitudinal ditch without producing perithecia, superficial perithecia. Or the formation of anamorphic synnemata, the phialides had a swollen basal portion, tapering abruptly into a narrow neck, conidia oval to fusiform, one-celled, often in chains [1,2]. In order to account for the phylogenetic diversity of isaria-like species and to segregate these isaria-like fungi from the *Akanthomyces* group, the genus *Samsoniella* was established [1]. *Isaria* Pers. was one of the oldest names for asexually typified genera in *Cordycipitaceae*; however, many entomogenous fungi morphologically similar to *Isaria* could be found distributed throughout *Hypocreales* [3]. In 2017, Kepler et al. revealed a polyphyletic distribution of *Isaria* species within *Cordycipitaceae*, proposed the rejection of *Isaria* and combined 11 species of *Isaria* into *Cordyceps* Fr., owing to the confusion surrounding the application of *Isaria* [4]. Two isolates of *I. farinosa* (CBS 240.32 and CBS 262.58) that remained genetically distant from CBS 111113 were renamed *S. alboaurantium* [1].

The species of *Samsoniella* have diverse biological characteristics. The genus currently contains 18 species (http://www.indexfungorum.org, accessed on 1 May 2022), among which *S. hepiali* is an essential medicinal fungus [1,2]. The chemical profile of *S. hepiali* is very similar to the profiles of *O**phiocordyceps*
*sinensis*, and recent studies show that *S. hepiali* performs various biological pharmacological activities such as anti-cancer, analgesic and hypoglycaemic activity, and is a good substitute for *O. sinensis* [5]. The related species of *S. hepiali* may have similar pharmacological activities. However, no research on other members of the genus has yet been reported. 

During surveys of entomopathogenic fungi from different regions in Yunnan Province, Guizhou Province of Southwestern China and Dole Province of Vietnam, five *Samsoniella* species were found and identified. Based on morphological evidence together with multigene (nr*SSU*, nr*LSU*, *tef-1α*, *rpb1* and *rpb2*) sequence analyses, it was shown that these five new *Samsoniella* species were distinguished from other species of the genus. They were named *S. coccinellidicola*, *S. farinospora*, *S. hania**na*, *S. pseudotortricidae* and *S. sinensis*. Furthermore, the morphological characteristics of 23 species in *Samsoniella*, comprising 5 novel species and 18 known taxa, were also compared. 

## 2. Materials and Methods

### 2.1. Sample Collection and Isolation

The majority of the specimens used in this study were collected from Yunnan Province in China. Some specimens were collected from the Chu Yang Sin National Park of Dole Province in Vietnam. The specimens were noted and photographed in the fields. The sample was placed in an ice box and brought to the laboratory for preservation at 4 °C. To obtain axenic cultures, the stromata or synnemata were removed from the insect bodies and divided into 3–4 segments, each 2 mm long. The segments were immersed in 30% H_2_O_2_ for 30 s and then soaked in sterilized water for 1 minute. After drying on sterilized filter paper, the segments were inoculated onto potato dextrose agar (PDA: fresh potato 200 g/L, dextrose 20 g/L, and agar 18 g/L) plates. The conidia of cordycipitoid fungi at the conidial masses were picked using an inoculating loop and spread on PDA plates containing 0.1 g/L streptomycin and 0.05 g/L tetracycline [2]. Pure cultures were incubated at room temperature (about 25 °C). After isolation into pure cultures, they were transplanted to a PDA slant and stored at 4 °C. The specimens were deposited in the Yunnan Herbal Herbarium (YHH) at the Institute of Herb Biotic Resources, Yunnan University. The strain was deposited at the Yunnan Fungal Culture Collection (YFCC) of the Institute of Herb Biotic Resources, Yunnan University.

### 2.2. Morphological Observations

For descriptions of the sexual morph, fruiting bodies were photographed and measured using an Olympus SZ61 (Tokyo, Japan) stereomicroscope. Stromata were sectioned at a thickness of ca. 40 µm with a freezing microtome and mounted in water or lactic acid cotton blue on a slide for microscopic studies and photomicrography. The micro-morphological characteristics of the fungi, such as the perithecia, asci and ascospores, were examined using Olympus CX40 (Tokyo, Japan) and BX53 (Tokyo, Japan) microscopes. The circular agar blocks, circa 5 mm in diameter, from a colony were removed and placed on new PDA plates to observe the colony morphology. The colonies on PDA plates were cultured at 25 °C for 2 weeks, and the colony characteristics (size, texture and colour) were photographed with a Canon 700D camera. To observe the asexual morphological characteristics (e.g., conidiophores, phialides and conidia), Olympus CX40 and BX53 microscopes were employed.

### 2.3. DNA Extraction, PCR and Sequencing for Nuclear Genes

Total genomic DNA was extracted from axenic living cultures using the MiniBEST Plant Genomic DNA Extraction Kit (TaKaRa, Beijing, China), following the manufacturer’s instructions. The nuclear ribosomal small subunit (nr*SSU*) was amplified with the primer pair nr*SSU*-CoF and nr*SSU*-CoR [6]. The nuclear ribosomal large subunit (nr*LSU*) was amplified with the primer pair LR5 and LR0R [7,8]. The translation elongation factor 1α (*tef-1α*) was amplified with the primers EF1α-EF and EF1α-ER [9,10]. The largest and second largest subunits of RNA polymerase II (*rpb1* and *rpb2*) were amplified with the primers RPB1-5′F and RPB1-5′R, RPB2-5′F and RPB2-5′R, respectively [9,10]. In this study, five nuclear gene loci of all the samples were amplified, and the primers used were shown in Table 1. The above five pairs were synthesized by Kunming Xiuqi Technology Co., Ltd. Each 50 µL PCR included 25 μL of 2 × Taq PCR Master Mix (Tiangen Biotech Co., Ltd., Beijing, China), 0.5 µL of each forward and reverse primer (10 μM), 1 μL of genomic DNA and 23 μL of sterilized distilled water. The polymerase chain reaction (PCR) assay was performed as described by Wang et al. [11]. The PCR products were separated by electrophoresis in 1.0% agarose gels, purified using a Gel Band Purification Kit (Bio Teke Co., Ltd., Beijing, China) and then sequenced on an automatic sequence analyser (BGI Co., Ltd., Shenzhen, China). When the PCR products could not be sequenced directly, cloning was performed using a TaKaRa PMDTM18-T vector system (TaKaRa Biotechnology Co., Ltd., Dalian, China).

### 2.4. Phylogenetic Analyses

Phylogenetic analyses were performed based on the nr*SSU*, nr*LSU*, *tef-1α*, *rpb1* and *rpb2* sequences. The DNA sequences generated in this study were submitted to GenBank. Reference sequences were downloaded from NCBI (http://www.ncbi.nlm.nih.gov/, accessed on 1 May 2022). The specimen information and GenBank accession numbers were provided in Table 2. The sequences were aligned using the Clustal X2.0 (developted by European Bioinformatics Institute, Cambridge, the United Kingdom) and MEGA v6.06 (developted by Tokyo Metropolitan University, Tokyo, Japan) software with manual adjustment [12,13]. The aligned sequences of five genes were concatenated after sequence alignment and specific processing according to Wang et al. [2]. Phylogenetic analyses were conducted using the Bayesian Inference (BI) and the Maximum Likelihood (ML) methods employing MrBayes v3.1.2 and RAxML 7.0.3 [14,15]. The BI analysis was run on MrBayes v3.1.2 for five million generations using a GTR + G + I model determined by the jModelTest version 2.1.4 (developted by The University of Vigo, Vigo, Spain) [16]. The GTR + I was selected as the optimal model for the ML analyses, with 1000 rapid bootstrap replicates performed on the five-gene datasets.

## 3. Results

### 3.1. Sequencing and Phylogenetic Analyses

The 92 taxa of eight genera—*Akanthomyces*, *Amphichorda*, *Beauveria*, *Blackwellomyces*, *Cordyceps*, *Samsoniella*, *Simplicillium* and *Trichoderma*—were used for the ML and BI phylogenetic analyses. Two Trichoderma strains (*Trichoderma deliquescens* ATCC 208838 and Trichoderma stercorarium ATCC 62321) were designated as the outgroup. The concatenated sequence dataset of the five genes consisted of 4642 bp of sequence data (1055 bp for nrSSU, 897 bp for nrLSU, 969 bp for tef-1α, 756 bp for rpb1 and 965 bp for rpb2). Both phylogenetic trees from the BI and ML analyses exhibited similar topologies that had seven recognized, statistically well-supported clades in Cordycipitaceae, designated as Akanthomyces, Amphichorda, Beauveria, Blackwellomyces, Cordyceps, Samsoniella and Simplicillium (Figure 1). Most of the well-resolved genera and lineages in Cordycipitaceae shared similar relationships with previous analyses [1,4,10]. The 12 samples of five undescribed species also clustered in the genus Samsoniella clade based on the phylogenetic analyses of the combined dataset and were clearly distinct from *S. hepiali* and 16 described species, viz., *S. alboaurantia*, *S. alpina*, *S. antleroides*, *S. aurantia*, *S. cardinalis*, *S. coleopterorum*, *S. cristata*, *S. hymenopterorum*, *S. inthanonensis*, *S. kunmingensis*, *S. lanmaoa*, *S. pseudogunii*, *S. pupicola*, *S. ramosa*, *S. tortricidae* and *S. yunnanensis* (Figure 1). Similarly, phylogenetic relationships between the genus Samsoniella and closely related species, based on multigene dataset (nrLSU, nrSSU, tef-1α, rpb1 and rpb2) (see Figure 2). Both phylogenetic trees from the BI and ML analyses exhibited similar topologies and the five undescribed species also clustered in the genus *Samsoniella* clade that were clearly distinct from S. hepiali and 16 described species.


**SYNOPTIC KEYS**



*Samsoniella*



*Samsoniella alboaurantium*

*Samsoniella alpina*

*Samsoniella antleroides*

*Samsoniella aurantia*

*Samsoniella cardinalis*

*Samsoniella coccinellidicola*

*Samsoniella coleopterorum*

*Samsoniella cristata*

*Samsoniella farinospora*

*Samsoniella haniana*

*Samsoniella hepiali*

*Samsoniella hymenopterorum*

*Samsoniella inthanonensis*

*Samsoniella kunmingensis*

*Samsoniella lanmaoa*

*Samsoniella lepidopterorum*

*Samsoniella pseudogunii*

*Samsoniella pseudotortricidae*

*Samsoniella pupicola*

*Samsoniella ramosa*

*Samsoniella sinensis*

*Samsoniella tortricidae*

*Samsoniella yunnanensis*



**Teleomorph characters**
 Insect host 1. *Lepidoptera* (pupa,larva) .................................................................................................3, 5, 8, 13, 14, 15, 18, 20, 22 Stromata 1. Numbera. Fasciculate ................................................................................................................................................3, 13, 20, 22b. Several ...................................................................................................................................................................5c. Solitary or two ........................................................................................................................................................8d. Solitary or Several ...................................................................................................................................................18e. Solitary ....................................................................................................................................................................14f. Two to five ................................................................................................................................................................15 2. Size (long)a. 10~20 mm ..................................................................................................................................................................... 5b. 15~40 mm .......................................................................................................................................................8, 14, 20c. 20~70 mm ..............................................................................................................................................3, 13, 15, 18, 22 3. Shapea. Cylindrical to clavate, branches .........................................................................................................................3, 13b. Cylindrical, unbranches .............................................................................................................................................5c. Cylindrical, unbranches or dichotomous ..............................................................................................................18d. Crista-like, much branched .......................................................................................................................................8e. Cylindrical to clavate, bifurcated ...........................................................................................................................14f. Palmately branched ..................................................................................................................................................15g. Fascicular, multi-branched, often confluent at the base .....................................................................................20h. Unbranched or dichotomous ...................................................................................................................................22 Fertile parts 1. Shapea. Clavate to fake-like ....................................................................................................................................................3b. Clavate ...............................................................................................................................................................5, 14, 15c. Clavate to subulate ....................................................................................................................................................18d. Crista-like or subulate .........................................................................................................................................8, 22e. Having no obvious boundary with stipes, with a tapering sterile part ...........................................................20 2. Coloura. Orange red ................................................................................................................................................................13b. Orange to orange red ................................................................................................................................................3c. Scarlet ..........................................................................................................................................................................5d. Reddish orange ........................................................................................................................................8, 14, 15, 18e. White to pale brown ..........................................................................................................................................20, 22 Perithecia 1. shapea. Superficial, ovoid ........................................................................................................................................................13b. Superficial, fusiform ................................................................................................................................................3c. Superficial, oblong-ovate to fusiform ...................................................................................................................5d. Superficial, narrowly ovoid ..................................................................................................................................8e. Superficial, narrowly ovoid to fusiform ....................................................................................14, 15, 18, 20, 22 2. Lengtha. 280~450 µm ................................................................................................................................................3, 14, 18, 20b. 330~470 µm ..........................................................................................................................................................15, 22c. 370~490 µm ........................................................................................................................................................5, 8, 13 3. Widtha. 110~210 µm ......................................................................................................................................... 14, 15, 18, 20b. 130~250 µm ................................................................................................................................................3, 5, 8, 22c. 200~260 µm ..............................................................................................................................................................13 Asci 1. Shapea. Cylindrical, eight-spored, hyaline ...........................................................................................3, 5, 8, 13, 14, 15, 22 2. Lengtha. 150~300 µm ............................................................................................................................................3, 13, 14, 22b. 160~360 µm .....................................................................................................................................................5, 8, 15 3. Widtha. 2~3 µm ................................................................................................................................................................3, 13b. 2.5~4 µm .................................................................................................................................................................22c. 3~5 µm .......................................................................................................................................................5, 8, 14, 15 Ascospores 1. Shapea. Bola-shaped, septate, central part fliform, terminal part narrowly fusiform .....................3, 5, 8, 13, 14, 15, 22 2. Lengtha. 110~185 µm ................................................................................................................................................................3b. 120~300 µm .....................................................................................................................................5, 8, 13, 14, 15, 22 3. Widtha. 0.5~1.0 µm ......................................................................................................................................................... 5, 13b. 0.8~1.5 µm ............................................................................................................................................3, 8, 14, 15, 22 
**Anamorph characters**
 Insect host 1. *Lepidoptera* (pupa, larva) .................................................................................1, 2, 4, 9, 10, 11, 16, 17, 19, 21, 232. *Dermaptera* ................................................................................................................................................................213. *Coleoptera* (Snout beetle) .....................................................................................................................................6, 74. *Hymenoptera* (bee) ....................................................................................................................................................125. *Araneae* .......................................................................................................................................................................9 Synnemata1. Present ......................................................................................................................................2, 4, 6, 10, 11, 21, 23Irregularly branched ................................................................................................................................2, 4, 6, 10, 21b. Branched or unbranched ........................................................................................................................................11c. Gregarious ................................................................................................................................................................23 2. Not observed ..............................................................................................................................1, 7, 9, 12, 16, 17, 19 Cultural characteristics on PDA1. Growth rate on PDA at 25 °C at 2 wka. Relatively rapid (>60 mm diam) ..........................................................................................................................12b. Moderate (30–60 mm diam) ...........................................................................................2, 6, 7, 9, 11, 16, 17, 21, 23c. Slow (<30 mm diam) ......................................................................................................................................4, 10, 19 2. Conidiophoresa. Biverticillate ............................................................................................................................................................2b. Solitary .............................................................................................................................................................11, 19c. Solitary or verticillate .........................................................................................................6, 7, 9, 10, 12, 16, 21, 23d. Verticillate .......................................................................................................................................................1, 4, 17 3. Phialides number in a whorlsa. 2–4 ..............................................................................................................................................................1, 4, 7, 9, 16b. 2–5 ...........................................................................................................................................................6, 10, 11, 21c. 2–7 .......................................................................................................................................................................2, 23b. 2–9 .....................................................................................................................................................................17, 19e. 3–4 .............................................................................................................................................................................12 4. Shape of conidiaa. fusiform or oval ............................................................................................................................2, 4, 6, 10, 11, 12, 23b. fusiform, ellipsoidal or subglobose ..........................................................................................................................7c. fusiform to subglobose ............................................................................................................................................16d. fusiform ..............................................................................................................................................................17, 19e. ellipsoidal to fusiform, sometimes lemon-shaped ...................................................................................................1f. oblong to cylindrical ..................................................................................................................................................9g. spherical, elliptical or fusiform .................................................................................................................................21 
**Key to**
**
*Samsoniella*
**
**species**
 Sexual state present .........................................................................................................................................................1Sexual state not observed ................................................................................................................................................5  1a. Stromata fasciculate ...................................................................................................................................................2  1b. Stromata not fasciculate ...........................................................................................................................................4  2a. Stromata cylindrical to clavate, branches ..........................................................................................................................3  2b. Stromata fascicular, multi-branched, often confluent at the base ..................................................................***S. ramosa***  2c. Stromata unbranched or dichotomous .........................................................................................................***S. tortricidae***  3a. Fertile parts clavate to fake-like, orange to orange red; Perithecia superficial, fusiform .......................***S. antleroides***  3b. Fertile parts clavate, orange red; Perithecia superficial, ovoid ..............................................................***S. inthanonensis***  4a. Stromata several, cylindrical, unbranches; Fertile parts clavate, scarlet .....................................................***S. cardinalis***  4b. Stromata solitary or two, crista-like, much branched; Fertile parts crista-like or subulate, reddish orange; Ascospores 155–290 × 1.0–1.3 μm ...................................................................................................................................***S. cristata***  4c. Stromata solitary or several, cylindrical, unbranches or dichotomous; Fertile parts clavate to subulate, reddish orange; No mature ascospores were observed ........................................................................................***S. pseudotortricidae***  4d. Stromata solitary, cylindrical to clavate, bifurcated; Fertile parts clavate, reddish orange; Ascospores 127–190 × 0.5–1.5 μm ..........................................................................................................................................................***S. kunmingensis***  4e. Stromata two to five, palmately, branched; Fertile parts clavate, reddish orange; Ascospores 135–260 × 0.9–1.4 μm .................................................................................................................................................................................***S. lanmaoa***  5a. Synnemata Present .................................................................................................................................................................6  5b. Synnemata not observed ......................................................................................................................................................11  6a. Synnemata irregularly branched ..........................................................................................................................................7  6b. Synnemata branched or unbranched; Conidiophores solitary, with phialides in whorls of two to five ...................................................................................................................................................................................***S. hepiali***  6c. Synnemata gregarious; Conidiophores solitary or verticillate, with phialides in whorls of two to seven .....................................................................................................................................................................***S. yunnanensis***  7a. Colonies on PDA at 25 °C at 2 wk growing moderate (30–60 mm diam) ......................................................................8  7b. Colonies on PDA at 25 °C at 2 wk growing slow (<30 mm diam) ................................................................................10  8a. Conidiophores biverticillate....................................................................................................................................***S. alpina***  8b. Conidiophores solitary or verticillate ................................................................................................................................9  9a. Conidia fusiform or oval, 1.8–3.0 × 1.3–2.0 μm .....................................................................................***S. coccinellidicola***  9b. Conidia spherical, elliptical or fusiform, 2.0–3.1 × 1.3–1.9 μm .........................................................................***S. sinensis***  10a. Conidiophores solitary or verticillate, with phialides in whorls of two to five; Conidia fusiform or oval, 2.3–3.7 × 1.2–2.8 μm ...................................................................................................................................................................***S. haniana***  10b. Conidiophores verticillate, with phialides in whorls of two to four; Conidia fusiform or oval, 2.5–3.5 × 1.5 μm .................................................................................................................................................................................***S. aurantia***  11a. Colonies on PDA at 25 °C at 2 wk growing relatively rapid (>60 mm diam) ..............................***S. hymenopterorum***  11b. Colonies on PDA at 25 °C at 2 wk growing moderate (30–60 mm diam) ..................................................................12  11c. Colonies on PDA at 25 °C at 2 wk growing slow (<30 mm diam) ...............................................................***S. pupicola***  12a. Conidiophores verticillate ...............................................................................................................................................13  12b. Conidiophores solitary or verticillate ............................................................................................................................14  13a. Have phialides in whorls of two to four; Conidia ellipsoidal to fusiform, sometimes lemon-shaped, 2.0–3.0 × 1.0–1.8 μm ..........................................................................................................................................................***S. alboaurantium***  13b. Have phialides in whorls of two to nine; Conidia fusiform, 2.8–3.2 × 1.7–2.1 μm .............................***S. pseudogunii***  14a. Conidia fusiform, ellipsoidal or subglobose, 1.7–2.5 × 1.2–1.8 μm ...................................................***S. coleopterorum***  14b. Conidia oblong to cylindrical, 1.6–2.8 × 0.6–1.2 μm ..................................................................................***S. farinospora***  14c. Conidia fusiform to subglobose, 2.0–2.5 × 1.2–2.0 μm .......................................................................***S. lepidopterorum***

### 3.2. Taxonomy

The key morphological characteristics that distinguish the current *Samsoniella* species were summarized in the literature (Table 3 and Table 4). Including the five new species, there were 23 species of *Samsoniella* involved in the current study, among which we compared 9 species of the sexual morphs in *Samsoniella* (Table 3) and 22 species of the asexual morphs in *Samsoniella* (Table 4).

***Samsoniella coccinellidicola*** H. Yu, Y. Wang & Z.Q. Wang, sp. nov. (Figure 3).

**MycoBank****:** MB 844383.

**Etymology:** “coccinellidicola” refers to the host (*Coleoptera*: *Coccinellidae*).

**Holotype:** China, Yunnan Province, Kunming City, Xishan Forest Park. On the *Coccinellidae* buried in soil, 12 August 2017, Yao Wang, (YHH 20178, holotype; YFCC 8772, ex-holotype living culture).

**Sexual morph****:** Undetermined.

**Asexual morph****:** Two synnemata arising from oval cocoons of insect host. Synnemata erect, irregularly branched, starting 2–2.5 mm above the oval cocoons of insect host, 15–25 × 0.8–1.2 mm, pale yellow, isaria-like morph producing a mass of conidia along the synnemata, powdery and floccose. Colonies on PDA fast-growing, 49–52 mm diameter in 14 days at 25 °C, white, cottony, sporulating abundantly, reverse white to pale yellow. Hyphae smooth-walled, branched, septate, hyaline, 0.7–2.1 µm wide. Conidiophores smooth-walled, cylindrical, solitary or verticillate, 4.8–15 × 1.0–1.9 µm. Phialides verticillate, usually in whorls of two to five, or solitary on hyphae, 6.0–14.1 µm long, basal portion cylindrical to narrowly lageniform, tapering gradually or abruptly toward the apex, from 1.0–2.0 µm wide (base) to 0.3–0.8 µm wide (apex). Conidia smooth and hyaline, fusiform or oval, one-celled, 1.8–3.0 × 1.3–2.0 µm, often in chains. Size and shape of phialides and conidia similar in culture and on natural substratum.

**Host:** Coccinellidae.

**Habit****at****:** On the adults of *Coccinellidae* sp. buried in soil.

**Distribution:** Currently only known in Kunming City, Yunnan Province, China.

**Other m****aterial examined:** China, Yunnan Province, Kunming City, Xishan Forest Park. On the *Coccinellidae* buried in soil, 12 August 2017, Yao Wang (YHH 20179; YFCC 8773, living culture).

**Notes****:** The phylogenetic analysis of five genes showed that *S. coccinellidicola* was closely related to *S**. pupicola*. Morphologically, the new species *S. coccinellidicola* was distinctly different from *S**. pupicola* due to its longer phialides (6.0–14.1 µm), smaller conidia (1.8–3.0 × 1.3–2.0 µm) and conidia shape. Moreover, *S**. coccinellidicola* was found to occur on an adult beetle (*Coleoptera*: *Coccinellidae*), while *S**. pupicola* was found on a Lepidopteran pupa. Based on the previous studies of cordycipitaceous isaria-like fungi as well as our study, there were two species of parasitic *Samsoniella* in the order *Coleoptera*, i.e., *S**. coccinellidicola* and *S. coleopterorum*. However, *S. coccinellidicola* was easily distinguished from *S. coleopterorum* by its longer phialides (6.0–14.1 µm).

***Samsoniella farinospora*** H. Yu, Y. Wang & Z.Q. Wang, sp. nov. (Figure 4).

**MycoBank****:** MB 844384.

**Etymology:** The species name refers to the farinose conidia covering the host.

**Holotype:** Vietnam, Dole Province, Chu Yang Sin National Park. On a spider on the back of fresh leaves, 22 October 2017, Hong Yu (YHH 20180, holotype; YFCC 8774, ex-holotype living culture).

**Sexual morph****:** Undetermined.

**Asexual morph****:** Mycosed hosts covered by dense white to lavender mycelia, produces numerous white, powdery conidia. Colonies on PDA fast-growing, 47–50 mm in diameter after 14 days at 25 °C, villiform, light yellow in the middle with a white edge, middle hyphae thickening, reverse light yellow. Hypha smooth-walled, hyaline, septate, 0.7–1.8 µm wide. Conidiophores smooth-walled, cylindrical, solitary or verticillate, 2.4–14.0 × 0.9–1.8 µm. Phialides verticillate, usually in whorls of two to four or solitary on hyphae, 3.0–13.5 µm long, basal portion cylindrical to narrowly lageniform, tapering gradually or abruptly toward the apex, from 0.6–1.6 µm wide (base). Conidia smooth and hyaline, oblong to cylindrical, one-celled, 1.6–2.8 × 0.6–1.2 µm, often in chains. Size and shape of phialides and conidia similar in culture and on natural substratum.

**Host:** Spider, larva of *Hepialus*.

**Habit****at****:** On a spider on the back of fresh leaves, with a larva of *Hepialus* clinging to fallen leaves.

**Distribution:** Currently only known in Chu Yang Sin National Park, Dole Province, Vietnam.

**Other m****aterial examined:** Vietnam, Dole Province, Chu Yang Sin National Park. On a larva of *Hepialus* clinging to fallen leaves, 26 October 2017, Hong Yu (YHH 20188; YFCC 9051, living culture).

**Notes****:** Morphologically, *S. farinospora* resembled the phylogenetically sister species *S. hepiali.* They had the same host, the Hepialid larva, and isaria-like asexual conidiogenous structures, producing synnemata with powdery conidia at the apex. However, *S. farinospora* was also found to occur on a spider. Parasitic *Samsoniella* species on spiders had rarely been reported. In addition, our morphological observation revealed a significant difference in conidia sizes between *S. farinospora* (1.6–2.8 × 0.6–1.2 µm) and *S. hepiali* (1.8–3.3 × 1.4–2.2 µm). Both the morphological study and phylogenetic analyses of combined nr*SSU*, nr*LSU*, *tef-1α*, *rpb1* and *rpb2* sequence data supported the idea that this fungus was a distinctive species in the genus of *Samsoniella*.

***Samsoniella hania******na*** H. Yu, Y. Wang & Z.Q. Wang, sp. nov. (Figure 5).

**MycoBank****:** MB 844385.

**Etymology:** The haniana was named after the Hani nationality, living in Yunnan.

**Holotype:** China, Yunnan Province, Yuanyang County, Xinjie Town, Duoyishuxia Village. On a pupa of *Lepidoptera* in cocoons buried in soil, 15 December 2021, Yao Wang (YHH 20175, holotype; YFCC 8769, ex-holotype living culture).

**Sexual morph****:** Undetermined.

**Asexual morph****:** Synnemata arising from every part of the body of the insect host. Synnemata erect, usually irregularly branched at the apex, 20–40 × 1–1.8 mm, pale orange. isaria-like morph producing a mass of conidia at the branch apex, powdery and floccose. Colonies derived from germinating conidia. Colonies on PDA growing well, 24–29 mm diameter in 14 days at 25 °C, white, cottony, sporulating abundantly, reverse light orange. Hyphae smooth-walled, branched, septate, hyaline, 0.8–2.8 µm wide. Conidiophores smooth-walled, cylindrical, solitary or verticillate, 3.8–10.2 × 1.1–2.9 µm. Phialides verticillate, usually in whorls of two to five, or solitary on hyphae, 5.4–12.1 µm long, basal portion cylindrical to narrowly lageniform, tapering gradually or abruptly toward the apex, from 1.2–2.9 µm wide (base) to 0.3–1.1 µm wide (apex). Conidia smooth and hyaline, fusiform or oval, one-celled, 2.3–3.7 × 1.2–2.8 µm, often in chains. Size and shape of phialides and conidia similar in culture and on natural substratum.

**Host:** Pupae of *Lepidoptera*.

**Habit****at****:** On the pupae of *Lepidoptera* in cocoons buried in soil.

**Distribution:** Currently only known in Yuanyang County, Yunnan Province, China; Puer City, Yunnan province, China.

**Other m****aterial examined:** China, Yunnan Province, Yuanyang County, Xinjie Town, Duoyishuxia Village. On a pupa of *Lepidoptera* in cocoons buried in soil, 15 December 2021, Yao Wang (YHH 20176; YFCC 8770, living culture); China, Yunnan province, Puer City, Simao District, Simao Gang Town, Dajiu Village. On a pupa of *Lepidoptera* in a cocoon buried in soil, 23 August 2021, Zhi-Qin Wang (YHH 20177; YFCC 8771, living culture).

**Notes****:** Phylogenetically, *S. hania**na* was identified as a *Samsoniella* species based on the phylogenetic analyses and was closely related to *S. pseudogunii* and *S. coleopterorum* (Figure 1). However, three samples of *S. hania**na* were clustered together with strong statistical support and formed a separate clade. Morphologically, *S. hania**na* differed from *S. pseudogunii* due to its several synnemata (usually irregularly branched at the apex) and oval conidia. *Samsoniella hania**na* was distinguished from *S. coleopterorum* with several synnemata (irregularly branched at the apex), longer phialides (5.4–12.1 µm) and larger conidia (2.3–3.7 × 1.2–2.8 µm).

***Samsoniella pseudotortricidae*** H. Yu, Y. Wang & Z.Q. Wang, sp. nov. (Figure 6).

**MycoBank****:** MB 844386.

**Etymology:** Referring to macromorphological resemblance of *S**. tortricidae* and *S. pseudotortricidae* but phylogenetically distinct.

**Holotype:** China, Yunnan Province, Kunming City, Wild Duck Lake Forest Park. On a pupa of *Lepidoptera* in cocoons buried in soil, 12 August 2017, Hong Yu (YHH 20174, holotype; YFCC 9052, ex-holotype living culture).

**Sexual morph****:** Stromata arising from insect cocoon, solitary to several, up to 20–65 mm long, unbranched or dichotomous. Stipes fleshly, flexuous, orange, cylindrical to clavate, 10–43 × 1.1–3.3 mm. Fertile parts reddish orange, clavate to subulate, lateral side usually have a longitudinal section without producing perithecia, 10–17 × 1.5–4.2 mm. Perithecia crowded, superficial, narrowly ovoid to fusiform, 285.7–313.2 × 149.2–154.9 µm. No mature asci or ascospores were observed.

**Asexual morph****:** isaria-like. Colonies on PDA grow well, 30–36 mm diameter in 14 days at 25 °C, white, cottony, sporulating abundantly, reverse light orange. Hyphae smooth-walled, branched, septate, hyaline, 1.1–1.5 µm wide. Conidiophores smooth-walled, cylindrical, solitary or verticillate, 6.6–26.5 × 1.1–2.5 µm. Phialides verticillate, in whorls of two to five, usually solitary on hyphae, 5.4–6.9 µm long, basal portion cylindrical to narrowly lageniform, tapering gradually or abruptly toward the apex, from 1.0–1.6 µm wide (base) to 0.5–0.8 µm wide (apex). Conidia smooth and hyaline, oblong, fusiform or oval, one-celled, 0.9–1.5 × 0.8–1.3 µm, often in chains.

**Host:** Pupae of *Lepidoptera*.

**Habit****at****:** On pupae of *Lepidoptera* in cocoons buried in soil.

**Distribution:** Currently only known in Kunming City, Yunnan Province, China.

**Other m****aterial examined:** China, Yunnan Province, Kunming City, Wild Duck Lake Forest Park. On a pupa of *Lepidoptera* in cocoons buried in soil, 12 August 2017, Hong Yu (YHH 20189, holotype; YFCC 9053, ex-holotype living culture).

**Notes****:***Samsoniella pseudotortricidae* was similar to its phylogenetically closely related species *S. tortricidae* in macromorphology. The stromata were both unbranched or dichotomous, both fertile parts were clavate to subulate, and reddish orange, and the lateral side usually had a longitudinal section without producing perithecia. However, *S. pseudotortricidae* was easily distinguished by its smaller ascus (285.7–313.2 × 149.2–154.9 µm), smaller phialides (5.4–6.9 × 1.0–1.6 µm) and smaller conidia (0.9–1.5 × 0.8–1.3 µm). It could be easily distinguished phylogenetically from *S. tortricidae*.

***Samsoniella sinensis*** H. Yu, Y. Wang & Z.Q. Wang, sp. nov. (Figure 7).

**MycoBank****:** MB 844387.

**Etymology:** Named after China (Yunnan and Guizhou provinces), where the species is distributed.

**Holotype:** China, Yunnan Province, Kunming City, Xishan Forest Park. On a larva of *Lepidoptera* clinging to fallen leaves, 12 August 2018, Hong Yu (YHH 20170, holotype; YFCC 8766, ex-holotype living culture).

**Sexual morph****:** Undetermined.

**Asexual morph****:** Synnemata arising from the host, 3.5–5 mm long, irregularly branched, conidia in abundance at the apex. Colonies fast-growing on PDA, 35–40 mm in 14 days at 25 °C, floccose, crater-shaped, white to pale pink, sporulating abundantly at the centrum, forming a white concentric ring. Reverse pale brown. Hyphae smooth-walled, branched, septate, hyaline, 1.3–3.1 µm wide. Conidiophores cylindrical, solitary or verticillate, 6.4–10.5 × 1.7–2.1 µm. Phialides verticillate, in whorls of two to five, sometimes solitary on hyphae, 5.6–9.3 µm long, basal portion cylindrical to narrowly lageniform, tapering gradually or abruptly toward the apex, from 1.5–2.1 µm wide (base) to 0.6–1.0 µm wide (apex). Conidia smooth and hyaline, spherical, elliptical or fusiform, one-celled, 2.0–3.1 × 1.3–1.9 µm, often in chains. Size and shape of phialides and conidia similar in culture and on natural substratum.

**Host:** Larvae of *Lepidoptera*, *Dermaptera*.

**Habit****at****:** On the larvae of *Lepidoptera* clinging to fallen leaves or on *Dermaptera* clinging to fallen leaves.

**Distribution:** Currently only known in Kunming City and Chuxiong City, Yunnan Province, China, and Guiyang City, Guizhou Province, China.

**Other m****aterial examined:** China, Yunnan Province, Kunming City, Kunming Wild Duck Lake Forest Park. On a pupa of *Dermaptera* clinging to fallen leaves, 13 August 2017, Yao Wang (YHH 20171; YFCC 8767, living culture); China, Yunnan Province, Chuxiong City, Zixi Mountain. On *Dermaptera* clinging to fallen leaves, 13 August 2016, Yao Wang (YHH 20172; YFCC 8768, living culture); China, Guizhou Province, Guiyang City. On a larva of *Lepidoptera*, 13 August 2017, Yao Wang (YHH 20173).

**Notes****:** Regarding phylogenetic relationships, *S. sinensis* formed a distinct lineage and was closely related to *S. hymenopterorum*. Morphologically, synnemata were observed in *S. sinensis*, and synnemata were not observed in *S. hymenopterorum*. The phialides of *S. sinensis* (5.6–9.3 µm) were shorter than those of *S. hymenopterorum* (6.5–10.6 µm). The conidia of *S. hymenopterorum* were fusiform to ovoid, 1.9–2.5 × 1.5–2.1 µm, but those of *S. sinensis* were spherical, elliptical or fusiform, 2.0–3.1 × 1.3–1.9 µm. *Samsoniella sinensis* was also easily distinguished from *S. hymenopterorum* by its host. *Samsoniella sinensis* was found to occur on *Lepidoptera* and *Dermaptera*, while *S. hymenopterorum* was only found to occur on *Hymenopterous* insects.

## 4. Discussion

The macromorphology and micromorphology of some *Samsoniella* species were very similar, and thus, the species were not easy to distinguish using only morphological characteristics [1,2]. In addition, *Samsoniella*, *Beauveria* and *Cordyceps* shared many similar morphological characteristics of sexual morphs, viz., fleshy stromata, red to orange colours, superficial perithecia, cylindrical asci with thickened ascus apex and usually cylindrical and multiseptate ascospores. *Samsoniella*, *Akanthomyces* and *Cordyceps* species produced similar isaria-like asexual conidiogenous structures, such as flask-shaped phialides produced in whorls and conidia with divergent chains [2]. It was more difficult to identify individual *Samsoniella* species. In the present study, a comprehensive morphological and phylogenetic investigation was conducted in most of the lineages of *Samsoniella.* The microscopic observations were compared with those for other known species in the genus, revealing some obvious differences, although the morphological features generally overlapped (Table 3 and Table 4). In comparison with other known species, parasitic *S. coccinellidicola* on adult beetles possessed relatively long phialides, *S. farinospora* had oblong to cylindrical conidia, *S. hania**na* produced larger conidia, *S. pseudotortricidae* had smaller conidia and *S. sinensis* produced a variety of shapes of conidia (viz., spherical, elliptical or fusiform). Based on the five-gene (nr*SSU*, nr*LSU*, *tef-1α*, *rpb1* and *rpb2*) dataset, molecular phylogenetic analyses also supported the existence of the five distinct species in the genus, emphasizing the importance of micromorphology and molecular identification (Figure 1).

*Samsoniella hepiali* has a great medical value due to its therapeutic effects in cardiovascular, respiratory, immunomodulatory, hyposexuality, hyperglycaemia and renal disorder conditions as well as its antitumor properties [2,33,34,35,36,37,38,39]. The Ministry of Health of the People’s Republic of China issued File No. 84 on 23 March 2001 and approved *S. hepiali* mycelia for use as a standalone preparation or a component of health foods (equivalent to dietary supplements in other countries) [40]. Thus, over 260 healthcare products have been developed with *S. hepiali* as a raw material in the global market, especially the Jinshuibao capsule [2]. To date, *S. hepiali* has been widely used as an edible and medicinal fungus, generating an impressive economic value of approximately RMB 10 billion a year in China [2]. It seemed to us that the related species of *S. hepiali* with similar genetic traits should have similar pharmacological activities. In this study, the *S. farinospora* strain YFCC 9051, isolated from a larva of *Hepialus*, and the other isolate, such as YFCC 8774, formed an independent clade apart from their allied species of *Samsoniella* and were further grouped with *S. hepiali* (see Figure 1). It was suggested that the two species should have a close genetic relationship. Morphologically, the strain YFCC 9051 was very similar to *S. hepiali*. They shared the same host of the hepialid larva, and both possessed an isaria-like asexual conidiogenous structure, producing synnemata with powdery conidia at the apex. Moreover, the main components in the mycelium of *S. farinospora* were similar to those in the mycelium of *S. hepiali*, involving adenosine, alkaloids, amino acids, ergosterol, mannitol, organic acids and polysaccharides (unpublished data). The strains of *S. farinospora* will be further determined to develop a raw material for healthcare products in future.

Previous studies of cordycipitaceous isaria-like fungi showed that species of *Samsoniella* were globally distributed generalist entomopathogen that were soilborne and had relatively complicated hosts, including *Lepidoptera* (*Hepialidae*, *Noctuidae*, *Limacodidae*, *Saturniidae* and *Tortricidae*), *Coleoptera* (*Curculionidae*), *Hymenoptera* (*Formicidae* and *Vespidae*) and two fungi (*O. sinensis* and *C. cicadae*) [1,2,41]. Here, an extension of the host range was identified, also including *Araneae*, *Dermaptera* and *Coccinellidae* of *Coleoptera*, as shown in Figure 1 and Table 2. Among the hosts of *Samsoniella* species, *Lepidoptera* was the major order (Table 2). Because of their broad host range and wide geographical distribution, some species of *Samsoniella* may have high potential for the interspecific transmission and biological control of pest insects. Additional research is needed to determine the effectiveness of isolates in the field.

## Figures and Tables

**Figure 1 jof-08-00747-f001:**
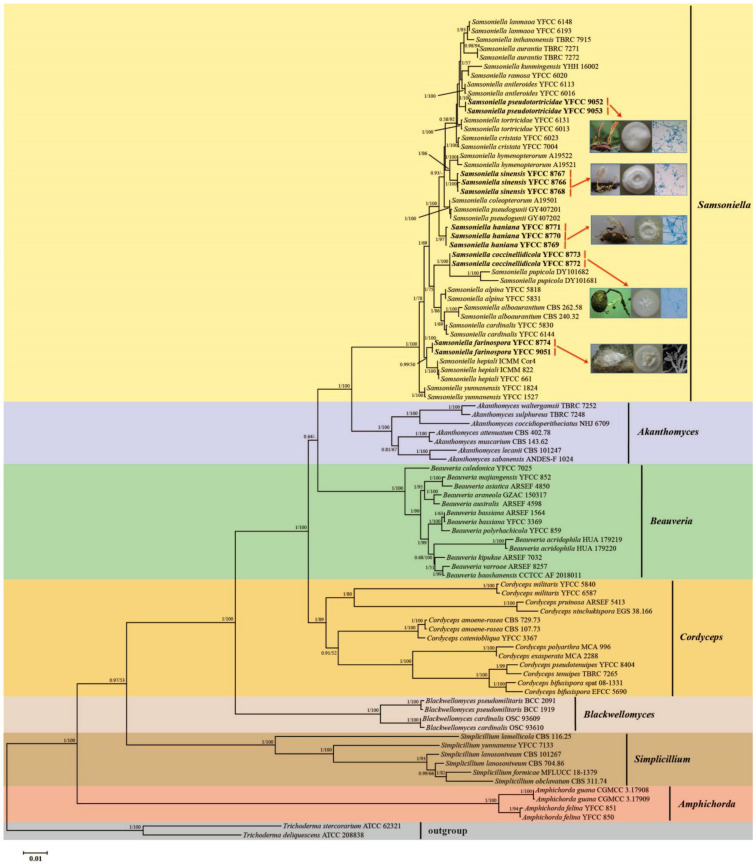
Phylogenetic tree of *Cordycipitaceae* inferred from multigene dataset (nr*LSU*, nr*SSU*, *tef-1α*, *rpb1* and *rpb2*) based on maximum likelihood (ML) and Bayesian inference (BI) analyses. Statistical support values greater than 50% are shown at the nodes for BI posterior probabilities/ML bootstrap proportions. Isolates in bold type are those analysed in this study.

**Figure 2 jof-08-00747-f002:**
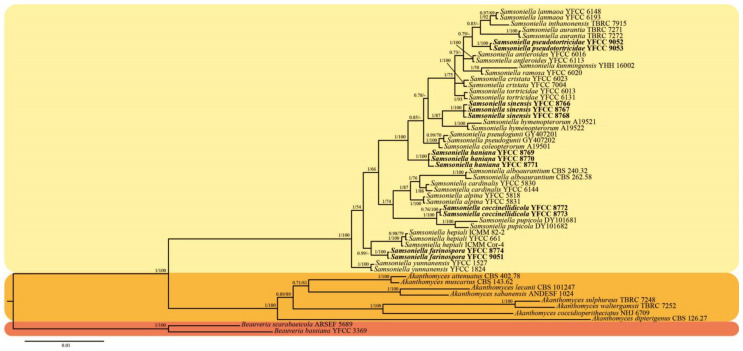
Phylogenetic relationships between the genus *Samsoniella* and closely related species, based on multigene dataset (nr*LSU*, nr*SSU*, *tef-1α*, *rpb1* and *rpb2*). Statistical support values greater than 50% are shown at the nodes for BI posterior probabilities/ML bootstrap proportions. Isolates in bold type are those analysed in this study.

**Figure 3 jof-08-00747-f003:**
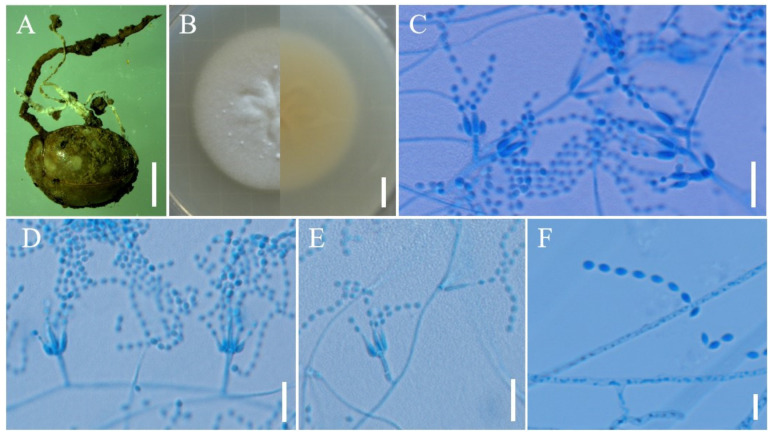
Morphology of *Samsoniella coccinellidicola.* (**A**) *Coccinellidae* infected by *S. coccinellidicola*. (**B**) Culture character on PDA medium. (**C–****E**) Conidiogenous cells (conidiophores, phialides) and conidia on PDA. (**F**) Conidia on PDA. Scale **A**: 5 mm; **B**: 10 mm; **C**–**E**: 10 µm; **F**: 5 µm.

**Figure 4 jof-08-00747-f004:**
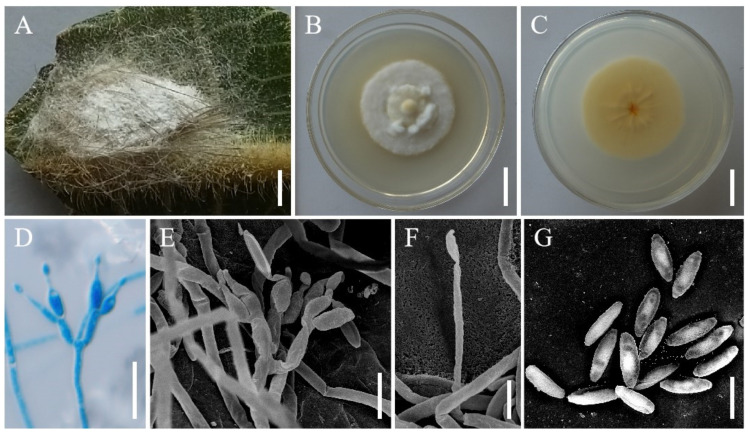
Morphology of *Samsoniella farinospora*. (**A**) Spider infected by *S. farinospora*. (**B**,**C**) Culture character on PDA medium. (**D**–**F**) Conidiogenous cells (conidiophores, phialides) and conidia on PDA. (**G**) Conidia on PDA. Scale **A**: 2 mm; **B**,**C**: 20 mm; **D**,**E**: 5 µm; **F**,**G**: 3 µm.

**Figure 5 jof-08-00747-f005:**
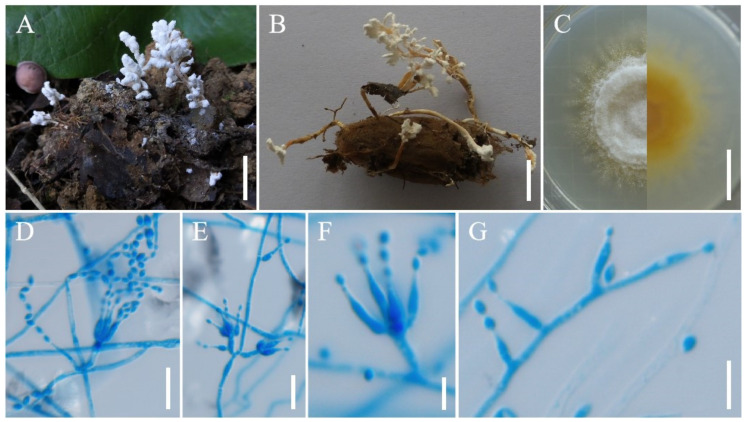
Morphology of *Samsoniella hania**na*. (**A**,**B**) Pupa of *Lepidoptera* infected by *S. hania**na*. (**C**) Culture character on PDA medium. (**D**–**G**) Conidiogenous cells (conidiophores, phialides) and conidia on PDA. Scale **A**,**B**: 10 mm; **C**: 20 mm; **D**,**E**: 10 µm; **F**,**G**: 5 µm.

**Figure 6 jof-08-00747-f006:**
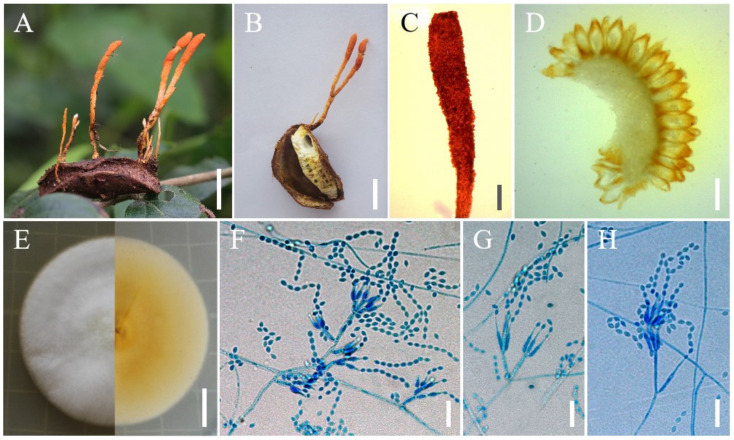
Morphology of *Samsoniella pseudotortricidae*. *(***A**,**B**) Stromata of fungus arising from lepidopteran pupa. (**C**) Fertile part. (**D**) Perithecia. **(****E****)** Culture character on PDA medium. (**F**–**H**) Conidiogenous cells (conidiophores, phialides) and conidia on PDA. Scale **A**,**B**: 10 mm; **C**: 1 mm; **D**: 300 µm; **E**: 1 cm; **F**–**H**: 10 µm.

**Figure 7 jof-08-00747-f007:**
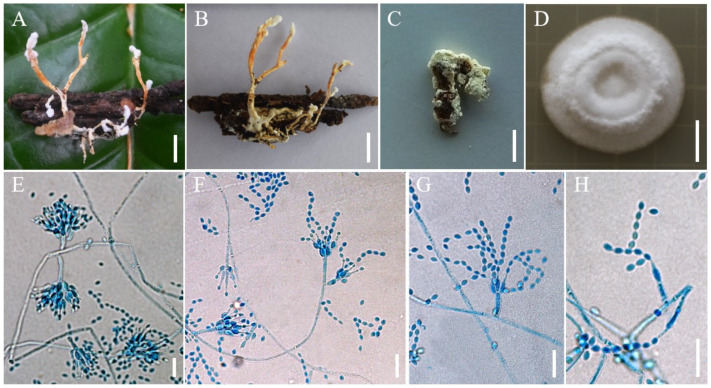
Morphology of *Samsoniella sinensis.* (**A**,**B**) Larva of *Lepidoptera* infected by *S. sinensis*. (**C**) *Dermaptera* infected by *S. sinensis*. (**D**) Culture character on PDA medium. (**E–H**) Conidiogenous cells (conidiophores, phialides) and conidia on PDA. Scale **A**,**B**: 5 mm; **C**: 3 mm; **D**: 10 mm; **E**–**H**: 10 µm.

**Table 1 jof-08-00747-t001:** PCR primers used in this study.

Gene	Primer	5′-Sequence-3′	References
nr*SSU*	nr*SSU*-CoF	TCTCAAAGATTAAGCCATGC	[6]
	nr*SSU*-CoR	TCACCAACGGAGACCTTG	
nr*LSU*	LR5	ATCCTGAGGGAAACTTC	[7,8]
	LR0R	GTACCCGCTGAACTTAAGC	
*tef-1α*	EF1α-EF	GCTCCYGGHCAYCGTGAYTTYAT	[9,10]
	EF1α-ER	ATGACACCRACRGCRACRGTYTG	
*rpb1*	RPB1-5′F	CAYCCWGGYTTYATCAAGAA	[9,10]
	RPB1-5′R	CCNGCDATNTCRTTRTCCATRTA	
*rpb2*	RPB2-5′F	CCCATRGCTTGTYYRCCCAT	[9,10]
	RPB2-5′R	GAYGAYMGWGATCAYTTYGG	

**Table 2 jof-08-00747-t002:** Names, voucher information, host and corresponding GenBank accession numbers of the taxa used in this study.

Taxon	Voucher Information	Host	GenBank Accession Number					References
			nr*SSU*	nr*LSU*	*tef-1α*	*rpb1*	*rpb2*	
*Akanthomyces attenuatus*	CBS 402.78	Leaf litter	AF339614	AF339565	EF468782	EF468888	EF468935	[10]
*coccidioperitheciatus*	NHJ 6709	*Araneae*	EU369110	EU369042	EU369025	EU369067	EU369086	[17]
*A. dipterigenus*	CBS 126.27	*Hemiptera*: *Monophlebidae*	AF339605	AF339556	KM283820	KR064300	KR064303	Unpublished
*A. lecanii*	CBS 101247	*Hemiptera*: *Coccidae*	AF339604	AF339555	DQ522359	DQ522407	DQ522466	[18]
*A. muscarius*	CBS 143.62	*Hemiptera*: *Aleyrodidae*	KM283774	KM283798	KM283821	KM283841	KM283863	Unpublished
*A. sabanensis*	ANDES-F 1024	*Hemiptera*: *Coccidae*	KC633251	KC875225	KC633266		KC633249	[19]
*A. sulphureus*	TBRC 7248	*Araneae*		MF140722	MF140843	MF140787	MF140812	[1]
*A. waltergamsii*	TBRC 7252	*Araneae*		MF140714	MF140834	MF140782	MF140806	[1]
*Amphichorda felina*	YFCC 850	Bird droppings	MW181774	MW173986	MW168227	MW168193	MW168210	[20]
*A. felina*	YFCC 851	Bird droppings	MW181775	MW173987	MW168228	MW168194	MW168211	[20]
*A. guana*	CGMCC 3.17908	Bat guano	KY883262	KU746711	KX855211	KY883202	KY883228	[21]
*A. guana*	CGMCC 3.17909	Bat guano	KY883263	KU746712	KX855212	KY883203		[21]
*Beauveria acridophila*	HUA 179219	*Orthoptera*: *Acrididae*		JQ895541	JQ958613	JX003857	JX003841	[22]
*B. acridophila*	HUA 179220	*Orthoptera*: *Acrididae*	JQ895527	JQ895536	JQ958614	JX003852	JX003842	[22]
*B. araneola*	GZAC 150317	*Araneae*			KT961699	KT961701		[23]
*B. asiatica*	ARSEF 4850	*Coleoptera*: *Cerambycidae*			AY531937	HQ880859	HQ880931	[24]
*B. australis*	ARSEF 4598	Soil			HQ880995	HQ880861	HQ880933	[24]
*B. bassiana*	YFCC 3369	*Coleoptera*: *Scarabaeidae*	MN576768	MN576824	MN576994	MN576884	MN576938	[2]
*B. baoshanensis*	CCTCC AF 2018011	*Coleoptera*: *Chrysomelidae*	MG642882	MG642840	MG642897	MG642854	MG642867	[25]
*B. caledonica*	YFCC 7025		MN576771	MN576827	MN576997	MN576887	MN576941	[2]
*B. kipukae*	ARSEF 7032	*Homoptera*: *Delphacidae*			HQ881005	HQ880875	HQ880947	[24]
*B. majiangensis*	YFCC 852	*Hemiptera*: *Pentatomidae*	MW181776	MW173988	MW168229	MW168195	MW168212	[20]
*B. polyrhachicola*	YFCC 859	*Hymenoptera*: *Formicidae*	MW181783	MW173995	MW168236	MW168202	MW168219	[20]
*B.scarabaeidicola*	ARSEF 5689	*Coleoptera*: *Scarabaeidae*	AF339574	AF339524	DQ522335	DQ522380	DQ522431	[20]
*B. varroae*	ARSEF 8257	*Coleoptera*: *Curculionidae*			HQ881002	HQ880872	HQ880944	[24]
*Blackwellomyces cardinalis*	OSC 93609	*Lepidoptera*: *Tineidae*	AY184973	AY184962	DQ522325	DQ522370	DQ522422	[18]
*B. cardinalis*	OSC 93610	*Lepidoptera*: *Tineidae*	AY184974	AY184963	EF469059	EF469088	EF469106	[18]
*B. pseudomilitaris*	BCC 1919	*Lepidoptera* (Larva)	MF416588	MF416534	MF416478		MF416440	[4]
*B. pseudomilitaris*	BCC 2091	*Lepidoptera* (Larva)	MF416589	MF416535	MF416479		MF416441	[4]
*Cordyceps amoene-rosea*	CBS 107.73	*Coleoptera* (Pupa)	AY526464	MF416550	MF416494	MF416651	MF416445	[26]
*C. amoene-rosea*	CBS 729.73	*Coleoptera*: *Nitidulidae*	MF416604	MF416551	MF416495	MF416652	MF416446	[26]
*C. bifusispora*	EFCC 5690	*Lepidoptera* (Pupa)	EF468952	EF468806	EF468746	EF468854	EF468909	[10]
*C. bifusispora*	spat 08-133.1	*Lepidoptera* (Pupa)	MF416577	MF416524	MF416469	MF416631	MF416434	[4]
*C. cateniobliqua*	YFCC 3367	*Coleoptera* adult	MN576765	MN576821	MN576991	MN576881	MN576935	[2]
*C. exasperata*	MCA 2288	*Lepidoptera* (Larva)	MF416592	MF416538	MF416482	MF416639		[4]
*C. militaris*	YFCC 6587	*Lepidoptera* (Pupa)	MN576762	MN576818	MN576988	MN576878	MN576932	[2]
*C. militaris*	YFCC 5840	*Lepidoptera* (Pupa)	MN576763	MN576819	MN576989	MN576879	MN576933	[2]
*C. ninchukispora*	EGS 38.166	Plant (*Beilschmiedia erythrophloia*)	EF468992	EF468847	EF468794	EF468901		[10]
*C. polyarthra*	MCA 996	*Lepidoptera*	MF416597	MF416543	MF416487	MF416644		[4]
*C. pruinosa*	ARSEF 5413	*Lepidoptera*: *Limacodidae*	AY184979	AY184968	DQ522351	DQ522397	DQ522451	[18]
*C. pseudotenuipes*	YFCC 8404	*Lepidoptera*	OL468559	OL468579	OL473527	OL739573	OL473538	[27]
*C. tenuipes*	TBRC 7265	*Lepidoptera* (Pupa)		MF140707	MF140827	MF140776	MF140800	[1]
*Samsoniella alboaurantium*	CBS 240.32	*Lepidoptera* (Pupa)	JF415958	JF415979	JF416019	JN049895	JF415999	[1]
*S. alboaurantium*	CBS 262.58	Soil			MF416497	MF416654	MF416448	[1]
*S. alpina*	YFCC 5818	*Hepialidae* (*Hepialus baimaensis*)	MN576753	MN576809	MN576979	MN576869	MN576923	[2]
*S. alpina*	YFCC 5831	*Hepialidae* (*Hepialus baimaensis*)	MN576754	MN576810	MN576980	MN576870	MN576924	[2]
*S. antleroides*	YFCC 6016	*Noctuidae* (Larvae)	MN576747	MN576803	MN576973	MN576863	MN576917	[2]
*S. antleroides*	YFCC 6113	*Noctuidae* (Larvae)	MN576748	MN576804	MN576974	MN576864	MN576918	[2]
*S. aurantia*	TBRC 7271	*Lepidoptera*		MF140728	MF140846	MF140791	MF140818	[1]
*S. aurantia*	TBRC 7272	*Lepidoptera*		MF140727	MF140845		MF140817	[1]
*S. cardinalis*	YFCC 5830	*Limacodidae* (Pupa)	MN576732	MN576788	MN576958	MN576848	MN576902	[2]
*S. cardinalis*	YFCC 6144	*Limacodidae* (Pupa)	MN576730	MN576786	MN576956	MN576846	MN576900	[2]
** *S. coccinellidicola* **	**YFCC 8772**	** *Coccinellidae* **	**ON563166**	**ON621670**	**ON676514**	**ON676502**	**ON568685**	**This study**
** *S. coccinellidicola* **	**YFCC 8773**	** *Coccinellidae* **	**ON563167**	**ON621671**	**ON676515**	**ON676503**	**ON568686**	**This study**
*S. coleopterorum*	A19501	*Curculionidae* (Snout beetle)			MN101586	MT642600	MN101585	[28]
*S. cristata*	YFCC 6021	*Saturniidae* (Pupa)	MN576735	MN576791	MN576961	MN576851	MN576905	[2]
*S. cristata*	YFCC 7004	*Saturniidae* (Pupa)	MN576737	MN576793	MN576963	MN576853	MN576907	[2]
** *S. farinospora* **	**YFCC 8774**	** *Araneae* ** **(Spider)**	**ON563168**	**ON621672**	**ON676516**	**ON676504**	**ON568687**	**This study**
** *S. farinospora* **	**YFCC 9051**	** *Lepidoptera* ** **:** ** *Hepialus* **	**ON563169**	**ON621673**	**ON676517**	**ON676505**	**ON568688**	**This study**
** *S. haniana* **	**YFCC 8769**	** *Lepidoptera* ** **(pupa)**	**ON563170**	**ON621674**	**ON676518**	**ON676506**	**ON568689**	**This study**
** *S. haniana* **	**YFCC 8770**	** *Lepidoptera* ** **(pupa)**	**ON563171**	**ON621675**	**ON676519**	**ON676507**	**ON568690**	**This study**
** *S. haniana* **	**YFCC 8771**	** *Lepidoptera* ** **(pupa)**	**ON563172**	**ON621676**	**ON676520**	**ON676508**	**ON568691**	**This study**
*S. hepiali*	ICMM 82-2	Fungi (*O. sinensis*)	MN576738	MN576794	MN576964	MN576854	MN576908	[2]
*S. hepiali*	YFCC 661	Fungi (*O. sinensis*)	MN576739	MN576795	MN576965	MN576855	MN576909	[2]
*S. hepiali*	Cor-4	Fungi (*O. sinensis*)	MN576743	MN576799	MN576969	MN576859	MN576913	[2]
*S. hymenopterorum*	A19521	*Vespidae* (Bee)			MN101588	MT642603	MT642604	[28]
*S. hymenopterorum*	A19522	*Vespidae* (Bee)			MN101591	MN101589	MN101590	[28]
*S. inthanonensis*	TBRC 7915	*Lepidoptera* (Pupa)		MF140725	MF140849	MF140790	MF140815	[1]
*S. kunmingensis*	YHH 16002	*Lepidoptera* (Pupa)	MN576746	MN576802	MN576972	MN576862	MN576916	[2]
*S. lanmaoa*	YFCC 6148	*Lepidoptera* (Pupa)	MN576733	MN576789	MN576959	MN576849	MN576903	[2]
*S. lanmaoa*	YFCC 6193	*Lepidoptera* (Pupa)	MN576734	MN576790	MN576960	MN576850	MN576904	[2]
*S. pseudogunii*	GY407201	*Lepidoptera* (Larvae)		MZ827010	MZ855233		MZ855239	[29]
*S. pseudogunii*	GY407202	*Lepidoptera* (Larvae)		MZ831865	MZ855234		MZ855240	[29]
** *S. pseudotortricidae* **	**YFCC 9052**	** *Lepidoptera* ** **(pupa)**	**ON563173**	**ON621677**	**ON676521**	**ON676509**	**ON568692**	**This study**
** *S. pseudotortricidae* **	**YFCC 9053**	** *Lepidoptera* ** **(pupa)**	**ON563174**	**ON621678**	**ON676522**	**ON676510**	**ON568693**	**This study**
*S. pupicola*	DY101681	*Lepidoptera* (Pupa)		MZ827009	MZ855231		MZ855237	[29]
*S. pupicola*	DY101682	*Lepidoptera* (Pupa)		MZ827635	MZ855232		MZ855238	[29]
*S. ramosa*	YFCC 6020	*Limacodidae* (Pupa)	MN576749	MN576805	MN576975	MN576865	MN576919	[2]
** *S. sinensis* **	**YFCC 8766**	** *Lepidoptera* ** **(Larvae)**	**ON563175**	**ON621679**	**ON676523**	**ON676511**	**ON568694**	**This study**
** *S. sinensis* **	**YFCC 8767**	** *Dermaptera* **	**ON563176**	**ON621680**	**ON676524**	**ON676512**	**ON568695**	**This study**
** *S. sinensis* **	**YFCC 8768**	** *Dermaptera* **	**ON563177**	**ON621681**	**ON676525**	**ON676513**	**ON568696**	**This study**
*S. tortricidae*	YFCC 6013	*Tortricidae* (Pupa)	MN576751	MN576807	MN576977	MN576867	MN576921	[2]
*S. tortricidae*	YFCC 6131	*Tortricidae* (Pupa)	MN576750	MN576806	MN576976	MN576866	MN576920	[2]
*S. yunnanensis*	YFCC 1527	Fungi (*Cordyceps cicadae*)	MN576756	MN576812	MN576982	MN576872	MN576926	[2]
*S. yunnanensis*	YFCC 1824	Fungi (*Cordyceps cicadae*)	MN576757	MN576813	MN576983	MN576873	MN576927	[2]
*Simplicillium formicae*	MFLUCC 18-1379	*Hymenoptera*: *Formicidae*	MK765046	MK766512	MK926451	MK882623		[30]
*S. lamellicola*	CBS 116.25	Fungi (*Agaricus bisporus*)	AF339601	AF339552	DQ522356	DQ522404	DQ522462	[18]
*S. lanosoniveum*	CBS 704.86	Fungi (*Hemileia vastatrix*)	AF339602	AF339553	DQ522358	DQ522406	DQ522464	[18]
*S. lanosoniveum*	CBS 101267	Fungi (*Hemileia vastatrix*)	AF339603	AF339554	DQ522357	DQ522405	DQ522463	[18]
*S. obclavatum*	CBS 311.74	Air above sugarcane field	AF339567	AF339517	EF468798			[10]
*S. yunnanense*	YFCC 7133	Fungi (*A. waltergamsii*)	MN576728	MN576784	MN576954	MN576844		[2]
*Trichoderma deliquescens*	ATCC 208838	On decorticated conifer wood	AF543768	AF543791	AF543781	AY489662	DQ522446	[31]
*T. stercorarium*	ATCC 62321	Cow dung	AF543769	AF543792	AF543782	AY489633	EF469103	[31]

Boldface: data generated in this study.

**Table 3 jof-08-00747-t003:** Comparison between the sexual morphs in *Samsoniella*.

Species	Stromata (mm)	Fertile Part (mm)	Perithecia (μm)	Asci (μm)	Ascospores (μm)	Reference
*Samsoniella antleroides*	fasciculate, antler-like, cylindrical to clavate, long 22.3–57.8, oblate terminal branches, long 4.6–26.2	clavate to fake-like, lateral sides have a longitudinal ditch without producing perithecia, 6.3–9.5 × 0.6–2.3	superficial, fusiform, 294–442 × 131–216	cylindrical, 8-spored, 160–248 × 2.1–2.7	bola-shaped, septate, 110–184 × 0.8–1.3	[2]
*S. cardinalis*	several, cylindrical, long 11.5–18.6	clavate, lateral sides have a longitudinal ditch without producing perithecia, 2.5–6.8 × 0.5–2.6	superficial, oblong-ovate to fusiform, 370–485 × 140–238	cylindrical, 8-spored, 163–320 × 3.2–4.3	bola-shaped, septate, 165–230 × 0.5–0.9	[2]
*S. cristata*	solitary or two, crista-like, long 25–40, much branched	crista-like or subulate, 3.1–18.5 × 0.9–8.0	superficial, narrowly ovoid, 370–485 × 150–245	cylindrical, 8-spored, 180–356 × 3.0–4.8	bola-shaped, septate, 155–290 × 1.0–1.3	[2]
*S. inthanonensis*	gregarious, cylindrical to clavate, long 20–50, 1–1.5 broad	8–15 long, 1.5–2 broad	superficial, ovoid, (380–)417.5–474.5(–500) × (150–)205–260(–265)	cylindrical, 8-spored, 300 × 2–2.5	bola-shaped, 3 or 4septate, 221.5–267 × 0.5–1	[1]
*S. kunmingensis*	solitary, cylindrical to clavate, long 23, bifurcated,	clavate, lateral sides usually have a longitudinal ditch without producing perithecia, 3.3–4.2 × 0.8–1.2	superficial, narrowly ovoid to fusiform, 330–395 × 110–185	cylindrical, 8-spored, 150–297 × 3.0–4.6	bola-shaped, septate, 127–190 × 0.8–1.5	[2]
** *S. pseudotortricidae* **	**solitary to several, long 20–65, unbranched or dichotomous**	**clavate to subulate, lateral side usually have a longitudinal section without producing perithecia, 10–17 × 1.5–4.2**	**superficial, narrowly ovoid to fusiform, 285.7–313.2 × 149.2–154.9**			**This study**
*S. lanmaoa*	two to five, orange, long 38–69, palmately branched	clavate, lateral sides usually have a longitudinal ditch without producing perithecia, 8.5–11.2 × 0.6–2.3	superficial, narrowly ovoid to fusiform, 360–467 × 124–210	cylindrical, 8-spored, 160–325 × 3.3–4.8	bola-shaped, septate, 135–260 × 0.9–1.4	[2]
*S. ramosa*	fascicular, 15–32 × 0.8–1.5, multi-branched, often confluent at the base	having no obvious boundary with stipes	superficial, narrowly ovoid to fusiform, 340–435 × 130–197			[2]
*S. tortricidae*	gregarious, long 25–60, unbranched or dichotomous	clavate to subulate, lateral side usually has a longitudinal section without producing perithecia, 5–15 × 1.2–2.3	superficial, narrowly ovoid to fusiform, 350–468 × 140–225	cylindrical, 8-spored, 170–285 × 2.8–4.0	bola-shaped, septate, 120–235 × 0.8–1.3	[2]

Boldface: data generated in this study.

**Table 4 jof-08-00747-t004:** Comparison between the asexual morphs in *Samsoniella*.

Species	Synnemata (mm)	Conidiophores (μm)	Phialides	Phialides Size (μm)	Conidia (μm)	References
*Samsoniella alboaurantium*		30–400 × 2–2.5		5–8 × 2, tapering fairly abruptly at the tip	ovate to lemon-shaped, 2.3–2.5(–3) × 1.5–1.8	[32]
*S. alpina*	irregularly branched, 3–20 long, cylindrical or clavate stipes with white powdery heads	3.1–6.5 × 1.6–2.8	verticillate on conidiophores, solitary or verticillate on hyphae	4.7–9.5 × 1.9–3.1, wide (apex) 0.5–1.1, basal portion cylindrical to narrowly lageniform, tapering abruptly toward the apex	fusiform or oval, 2.0–3.1 × 1.3–2.1	[2]
*S. antleroides*		3.5–9.7 × 1.3–3.2	verticillate, in whorls of 2 to 9, sometimes solitary on hyphae	3.5–16.3 × 1.7–2.9, wide (apex) 0.5–1.0, basal portion cylindrical to narrowly lageniform, tapering abruptly toward the apex	fusiform or oval, 2.3–3.5 × 1.6–2.5	[2]
*S. aurantia*	irregularly branched starting 15–40 above the ground and continuously to the apex, 25–75 × 1–1.5		verticillate, in whorls of 2 to 4	(5–)5.5–8.5(–13) × 2–3, basal portion cylindrical to ellipsoidal, neck 2–4 × 1	fusiform, (2–)2.5–3.5(–4) × (1–)1.5(–2)	[1]
*S. cardinalis*		3.1–9.5 × 1.3–2.0	verticillate, in whorls of 2 to 5, sometimes solitary on hyphae	4.1–43.5 × 1.3–2.4, wide (apex) 0.6–1.2, basal portion cylindrical to narrowly lageniform, tapering gradually or abruptly toward the apex	fusiform or oval, 2.4–3.2 × 1.4–2.2	[2]
** *S. coccinellidicola* **	**I** **rregularly branched, starting 2–2.5 above the cocoons of insect host, 15–25 × 0.8–1.2**	**4.8–15 × 1.0–1.9**	**verticillate, usually in whorls of 2 to 5, or solitary on hyphae**	**6.0–14.1 × 1.0–2.0 wide (apex) 0.3–0.8, basal portion cylindrical to narrowly lageniform, tapering gradually or abruptly toward the apex**	**fusiform or oval, 1.8–3.0 × 1.3–2.0**	**This study**
*S. coleopterorum*			verticillate, in whorls of 2 to 4	5.4–9.7 × 1.2–1.8, a cylindrical to ellipsoidal basal portion, tapering into a short distinct neck	fusiform, ellipsoidal or subglobose, 1.7–2.5 × 1.2–1.8	[28]
*S. cristata*		3.6–11.5 × 1.7–2.5	verticillate, in whorls of 2 to 5, usually solitary on hyphae	4.5–23.2 × 1.6–2.7, wide (apex) 0.5–1.1, basal portion cylindrical to narrowly lageniform, tapering gradually or abruptly toward the apex	fusiform or oval, 2.4–3.2 × 1.6–2.3	[2]
** *S. farinospora* **		**2.4–14.0 × 0.9–1.8**	**verticillate, usually in whorls of 2 to 4, or solitary on hyphae**	**3.0–13.5 × 0.6–1.6, basal portion cylindrical to narrowly lageniform, tapering gradually or abruptly toward the apex**	**oblong to cylindrical, 1.6–2.8 × 0.6–1.2**	**This study**
** *S. haniana* **	**usually unbranched or irregularly branched at the apex, 20–40 × 1–1.8**	**3.8–10.2 × 1.1–2.9**	**verticillate, usually in whorls of 2 to 5, or solitary on hyphae**	**5.4–12.1 × 1.2–2.9, wide (apex) 0.3–1.1 basal portion cylindrical to narrowly lageniform, tapering gradually or abruptly toward the apex**	**fusiform or oval** **,** **2.3–3.7 × 1.2–2.8**	**This study**
*S. hepiali*	branched or unbranched, 5–41 long	4.0–7.6 × 1.4–2.2	verticillate, in whorls of 2 to 5, solitary or opposite on hyphae	3.5–13.6 × 1.3–2.1, wide (apex) 0.5–1.0, basal portion cylindrical to narrowly lageniform, tapering gradually or abruptly toward the apex	fusiform or oval, 1.8–3.3 × 1.4–2.2	[2]
*S. hymenopterorum*			Verticillate, in whorls of 3 to 4	6.5–10.6 × 1.2–2.0, a cylindrical basal portion, tapering to a distinct neck	fusiform to ovoid, 1.9–2.5 × 1.5–2.1	[28]
*S. inthanonensis*			verticillate in whorls of 2 to 5, sometimes solitary on hyphae	(4–)6.5–10(–12) × (1–)1.5–2(–3), cylindrical basal portion, tapering into a long neck, (1–)2.5(–4) × 0.5–1	short fusiform, (2–)3(–3.5) × 1.5–2	[1]
*S. lanmaoa*		3.8–13.3 × 1.5–2.1	verticillate, in whorls of 2 to 6, usually solitary on hyphae	3.5–20.7 × 1.7–2.6, wide (apex) 0.5–1.1, basal portion cylindrical to narrowly lageniform, tapering gradually or abruptly toward the apex	fusiform or oval, 1.9–2.7 × 1.4–2.0	[2]
*S. lepidopterorum*			Verticillate, in whorls of 2 to 4	5.2–8.5 (–13.1) × 1.1–1.7, ellipsoidal basal portion, tapering into a distinct neck	fusiform to subglobose, 2.0–2.5 × 1.2–2.0	[28]
** *S. pseudotortricidae* **		**6.6–26.5 × 1.1–2.5**	**verticillate, in whorls of 2 to 5, usually solitary on hyphae**	**5.4–6.9 × 1.0–1.6, wide (apex) 0.5–0.8 basal portion cylindrical to narrowly lageniform, tapering gradually or abruptly toward the apex**	**fusiform or oval, 0.9–1.5 × 0.8–1.3**	**This study**
*S. pseudogunnii*			solitary or in whorls of 2 to 9	6.8–11.0 × 2.2–2.4, cylindrical basal portion, tapering into a short distinct neck	fusiform 2.8–3.2 × 1.7–2.1	[29]
*S. pupicola*			solitary or in whorls of 2 to 9	7.0–9.2 × 2.5–3.3, a cylindrical basal portion, tapering into a short distinct neck	fusiform, 2.5–3.3 × 2.2–2.6	[29]
*S. ramosa*		4.3–10.5 × 1.3–2.4	verticillate, in whorls of 2 to 6, usually solitary on hyphae	5.3–14.6 × 1.3–2.8, wide (apex) 0.6–1.2, basal portion cylindrical to narrowly lageniform, tapering gradually or abruptly toward the apex	fusiform or oval, 2.0–3.6 × 1.5–2.2	[2]
** *S. sinensis* **	**3.5** **–** **5** **long** **, branched, conidia in abundance at the apex.**	**6.4–10.5 × 1.7–2.1**	**verticillate, in whorls of 2 to 5, sometimes solitary on hyphae**	**5.6–9.3 × 1.5–2.1, wide (apex) 0.6–1.0 basal portion cylindrical to narrowly lageniform, tapering gradually or abruptly toward the apex**	**spherical, elliptical or fusiform, 2.0** **–** **3.1 × 1.3–1.9**	**This study**
*S. tortricidae*		4.2–12.5 × 1.4–2.4	verticillate, in whorls of 2 to 5, usually solitary on hyphae	3.6–42.4 × 1.1–2.6, wide (apex) 0.4–0.9, basal portion cylindrical to narrowly lageniform, tapering gradually or abruptly toward the apex	fusiform or oval, 2.1–3.0 × 1.3–1.7	[2]
*S. yunnanensis*	gregarious, flexuous, fleshy, 4–18 long, with terminal branches of 3–7 × 1.0–2.0	4.2–23.5 × 1.4–2.3	verticillate, in whorls of 2 to 7, usually solitary on hyphae	4.5–11.6 × 1.2–2.4, wide (apex) 0.6–1.0, basal portion cylindrical to narrowly lageniform, tapering gradually or abruptly toward the apex	fusiform or oval, 2.0–3.3 × 1.1–2.2	[2]

Boldface: data generated in this study.

## Data Availability

Publicly available datasets were analyzed in this study. This data can be found here: http://www.ncbi.nlm.nih.gov, accessed on 1 May 2022.

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
