# Peer review of "Morphological and Phylogenetic Characterization Reveals Five New Species of Samsoniella (Cordycipitaceae, Hypocreales)"

_jof, 2022, doi:10.3390/jof8070747_

Round 1

Reviewer 1 Report

This paper is a relatively well-executed study describing five new species in Samsoniella mainly based on molecular phylogenetic analyses. It is a good contribution to our understanding of the genus Samsoniella in Cordycipitaceae. However, the paper in its current form needs to be thoroughly improved. I suggest the authors to give the paper to an English editing service to improve the quality. There are so many typos and some sentences need to be rephrased as I do not understand what the authors mean by some of their statements. The table need to be revised and made sure it fits the page and can read easily. There is no clear distinction between the species in some of the rows. The long GenBank numbers are also unsightly when they are split.

 A key to the genus would be very important to help mycologists also identify what they have. This could be added after the discussion. The photos of the phialides from the slide preparations are not so clear and sharp. Please replace them with better photos.

 My edits are in the attached pdf.

Author Response

Response to Reviewer 1 Comments

Point 1: There are some formatting, word spelling and grammar problems.

Response 1:  As for the format, spelling and grammar, we have made corrections according to the reviewer's suggestions.

Point 2: I suggest to do a Key to Samsoniella species at the end of the discussion. make a key to the species that includes both teleomorphic and anamorphic characters

Response 2: Thanks very much for the reviewer’s suggestion. We have added a Key to Samsoniella species at the results to the manuscript.

Point 3: there are only 7 genera in Cordycipitaceae in the tree, Trichoderma is not in Cordycipitaceae

Response 3: Thanks very much for the reviewer’s suggestion. We have made correction according to the reviewer’s suggestions.

Point 4: replace with clearer and sharper photos.

Response 4: We have provided as good microscopic photos as possible.

Point 5: However, the paper in its current form needs to be thoroughly improved. I suggest the authors to give the paper to an English editing service to improve the quality. There are so many typos and some sentences need to be rephrased as I do not understand what the authors mean by some of their statements.

Response 5: Our manuscript had English language editing by MDPI.

Reviewer 2 Report

Dear Authors

This paper is a good work that has interesting information with description of five new Samsoniella species. You will see that I have included some comments/suggestions directly using the PDF file. 

In addition, if you could prepare better microscopic photos using DIC option. It would be excellent.

Author Response

Response to Reviewer 2 Comments

Point 1: There are some formatting problems.

Response 1: As for the format, we have made correction according to the reviewer’s suggestions.

Point 2: Pleas also add the culture ex-type to all new species; It is better to move the information of the type to the "Holotype" part. Please do this for all the new species depending on the journal format.

Response 2: Thanks very much for the reviewer’s suggestion. We have made correction according to the reviewer’s suggestions.

Point 3: In addition, if you could prepare better microscopic photos using DIC option. It would be excellent.

Response 3: We have provided as good microscopic photos as possible.

Reviewer 3 Report

This is an interesting and well illustrated contribution.

All observations and comments are marked in the attached pdf document.

English revision required.

Table Editing

Consider whether phylogenetic analysis requires all 92 samples.

Revise some terminology in the descriptions of anamorphs and teleomorphs.

Unify upper and lower case subtitles.

Round 2

Reviewer 1 Report

I have seen the improved version of the manuscript and have only two suggestions for improvement. (1) In the Key to Samsoniella species it would be better to remove the number beside the species names (page 12-13 in lines 341-381 since it refers to the same thing, e.g., 2b. Stromata fascicular, multi-branched, often confluent at the base ................................20 S. ramosa 

(2) In the Figures please include information if the conidiogenous structures and conidia were from the culture or from the host. It would be best to have photos of conidiogenous cells and conidia from the host and from PDA culture.
